# Phylogenomic Analyses of Three Distinct Lineages Uniting *Staphylococcus cohnii* and *Staphylococcus urealyticus* from Diverse Hosts

**DOI:** 10.3390/microorganisms12081549

**Published:** 2024-07-29

**Authors:** L. Caroline House, Amer Hasan, Andi Asnayanti, Adnan A. K. Alrubaye, Jeff Pummill, Douglas Rhoads

**Affiliations:** 1John Brown University, Siloam Springs, AR 72761, USA; carolinelouisa77@gmail.com; 2Cell and Molecular Biology Pogram, University of Arkansas, Fayetteville, AR 72703, USA; amer.a@covm.uobaghdad.edu.iq (A.H.); aasnayan@uark.edu (A.A.); aakhalaf@uark.edu (A.A.K.A.); jpummil@uark.edu (J.P.); 3Department of Veterinary Public Health, College of Veterinary Medicine, University of Baghdad, Baghdad P.O. Box 1417, Iraq; 4Department of Poultry Science, University of Arkansas, Fayetteville, AR 72703, USA; 5Arkansas High Performance Computing Center, University of Arkansas, Fayetteville, AR 72703, USA

**Keywords:** osteomyelitis, horizontal transfer, genome evolution, *Staphylococcus*, bioinformatics

## Abstract

We sequenced and assembled genomes for 17 isolates of *Staphylococcus cohnii* isolated from osteomyelitis lesions in young broilers from two separate experiments where we induced lameness using a hybrid wire-litter flooring system. Whole genome comparisons using three different methods support a close relationship of genomes from both *S. cohnii* and *Staphylococcus urealyticus*. The data support three different lineages, which we designated as Lineage 1, Lineage 2, and Lineage 3, uniting these two species within an evolving complex. We present evidence for horizontal transfer between lineages of genomic regions from 50–440 kbp. The transfer of a 186 kbp region from Lineage 1 to Lineage 2 appears to have generated Lineage 3. Human-associated isolates appear to be limited to Lineages 2 and 3 but Lineage 2 appears to contain a higher number of human pathogenic isolates. The chicken isolates from our lameness trials included genomically diverse isolates from both Lineage 1 and 2, and isolates from both lineages were obtained from osteomyelitis lesions of individual birds. Our results expand the diversity of Staphylococci associated with osteomyelitis in poultry and suggest a high diversity in the microbiome of day-old chicks. Our data also support a reevaluation and unification of the taxonomic classifications of *S. cohnii* and *S. urealyticus*.

## 1. Introduction

The Staphylococcus genus of Gram-positive bacteria comprises species identified as normal flora and opportunistic pathogens, isolated from a wide variety of animals [1,2,3,4]. Some Staphylococcus species have been associated with specific hosts. Our research group has been exploring the conditions leading to lameness outbreaks in the broiler industry. Lameness typically affects 1–2% of broilers, with Bacterial Chondronecrosis with Osteomyelitis (BCO) contributing about half of these cases [5,6,7]. However, flocks can experience lameness outbreaks that spread through a facility, affecting 10–15% of the flock, where the majority of cases are BCO. One of the primary incidents that can contribute to an outbreak is a significant stress event (heat, respiratory infection, etc.) and we have employed growth on a suspended wire-floor as our primary source of stress [6,7,8,9]. Historically, *Staphylococcus aureus* has been most commonly associated with BCO and systemic infections in chickens [10,11,12,13,14,15,16,17,18]. More recently, *Staphylococcus agnetis* has also been found infecting poultry [19,20,21] Previously, *S. agnetis* had been primarily associated with sub-clinical mastitis in cattle [22,23,24], but has also now been reported in skin infections in camelids [25], cattle [26], and poultry [20,27]. We have characterized isolates of *S. agnetis* obtained from femoral and tibial head necrosis lesions from broilers reared on wire-flooring [19]. One particular isolate is highly virulent when administered in a single dose to young broilers, and can spread from infected birds through the air to induce lameness in other birds in the same facility [28]. Whole genome comparisons show that the chicken isolates are closely related to cattle isolates [21]. Pangenome analyses suggest that changes in host and tissue tropism of *S. agnetis* may be driven by horizontal transfer of mobile genomic elements from *S. aureus* [29]. In both *S. aureus* and *S. agnetis,* host preference appears to be facilitated by horizontal gene transfer [16,21,29,30].

Recently, we have developed a new hybrid system for inducing lameness where birds are raised on wire flooring to induce lameness and the infection spreads to other chickens in the same facility raised on standard litter flooring [31]. This system best models the infectious spread of BCO in broiler facilities. Additionally, we do not control the source of the infection, which can derive from the environment or the bird’s own microbiome. Over the course of two lameness trials in Fall 2020 and Spring 2021 using the hybrid wire/litter floor model, we recovered 77 distinct cultures from femoral or tibial head necrosis lesions, 35 of which were identified as *Staphylococcus cohnii*. This particular species has been primarily associated with infections in mammals, and more specifically humans [32,33,34], with very few reported cases of pathogenesis in poultry [35]. *S. cohnii* has been described as a species complex containing three recognizable clades [32,33], including two subspecies (*S. cohnii* subsp. *cohnii* and *S. cohnii* subsp. *urealyticus*). Whole genome analyses have led some to propose splitting the complex into two species [33] Given the paucity of publications, or genome assemblies, for *S. cohnii* infecting chickens, and the high-frequency of isolation in two successive BCO challenge experiments, we chose to examine the genomic diversity of our BCO isolates of *S. cohnii*. Standard microbiological techniques suggested a diverse population, and whole genome comparisons expand the available genomes for elucidating the evolution of this taxonomic group and our understanding of the phylogenetic evolution of pathogenesis and host tropism. Our results also contribute to a better understanding of the etiology of BCO outbreaks in the broiler industry.

## 2. Materials and Methods

### 2.1. Media

Media included: CHROMagar Orientation (CO) and CHROMagar *Staphylococcus* (CS) (DRG International, Inc., Springfield, NJ, USA) and tryptic soy broth (TSB) (Difco Laboratories, Franklin Lakes, NJ, USA). Difco bacteriological agar was added at 1.5% for solidification, when necessary.

### 2.2. S. cohnii Isolates

*S. cohnii* isolates were part of a collection of isolates obtained from swabs of femoral head necrosis lesions from Experiment 29 (fall 2020) and Experiment 30 (spring 2021). Details of the animal work have been published elsewhere [31]. Briefly, there were two pens of broiler breeder male chicks raised on wire flooring to induce lameness. Separately in the same facility, broiler breeder males were raised on standard litter flooring. Around 31 days of age, lameness developed in the birds on wire-flooring and the infection spreads to the birds on litter flooring. At 56 days of age, birds were euthanized and the femurs aseptically exposed for diagnosis of femoral head necrosis. Sterile swabs were used to sample the lesions then streaked on CO plates. Colony morphology and color were used for initial discrimination. Individual colonies were streak purified on new CO plates and PCR sequenced for the 16S v1-5 regions for species identification [19,36]. Cultures were stored at −80 °C in 40% glycerol 50% TSB, and at 4 °C on TSB agar slants.

PCR-ready DNA was extracted using NaOH on 20 µL aliquots from overnight cultures followed by 4-fold dilution as described [37]. Species and lineage specific qPCR (quantitative PCR) was in triplicates of 20 µL in 1x Taq Buffer (50 mM Tris pH 8.3, 1.25 mM MgCl_2_, 300 ng/µL BSA), 0.2 mM dNTPs, 0.25 µM primers, 1 X EvaGreen^®^ Dye (Biotium, Fremont, CA, USA), 2 µL of extracted DNA, and 4 U Taq Polymerase in 96 well plates in a CFX96 Touch Real-Time PCR Detection System (Bio-Rad Laboratories, Inc., Hercules, CA, USA). Cycle parameters were: 90 °C for 45 s, 10 cycles of 90 °C for 15 s, anneal temperature (see below) for 15 s, and 72 °C for 60 s; followed by 30 cycles with the same parameters with a plate read after each cycle. The products were then subjected to high resolution melt (HRM) analysis to verify that any amplification signal was not artifactual, consisting of 72 °C for 180 s, 90 °C for 60 s, 70 °C for 120 s, followed by a 70 °C to 90 °C melt with steps of 0.1 °C for 5 s and plate read. The amplification and melt curves were analyzed using CFX Manager v 3.1 (Bio-Rad Laboratories). Staphylococcal species verification targeted the pfbA gene [38]; for *S. cohnii* the primers were forward: 5′-GCACTCGAATKGCATCTTTAGAC-3′; reverse: 5′-GGGCAGATACACGYGCGATTC-3′; with anneal temperature 60 °C. *S. cohnii* Lineage 1 specific primers (forward: 5′-CTCGTACAACTTAAACCTAGTATAG-3′; reverse: 5′-GTTTAATTTGTGTGTGTAAATCTTTTTCGT-3′) targeted the gene for der, a GTPase that plays an essential role in the late steps of ribosome biogenesis. Lineage 2 specific primers (forward: 5′-CACATATGGCATTAATCGTTACATTGA-3′; reverse: 5′-CTTTACTAATTAAAAGTTGATKAACAAAATGG-3′) targeted the gene for Nat, an arylamine N-acetyltransferase. Anneal temperature for lineage-specific primers was 59 °C.

### 2.3. Genome Sequence and Assembly

Total DNA was isolated by the method of Dyer and Iandolo [39] as modified [21]. Briefly, overnight 2 mL TSB cultures were diluted into 20 mL of fresh TSB and grown 2 h at 37 °C to log phase before treatment with lysostaphin, followed by standard SDS-lysis, organic extraction, RNase/protease digestion, and ethanol precipitation. The final purification was performed using NanoSep 100 k Omega spin cups (Stock#: OD100C33, Pall Corporation, Port Washington, NY, USA) according to the manufacturer’s instructions. Purified total bacterial DNAs were submitted to SeqCenter (Pittsburgh, PA, USA) for 2 × 151 paired-end sequencing on an Illumina NextSeq 2000 (Illumina Corp., San Diego, CA, USA). Sequence data were uploaded to BV-BRC [40] for processing through the Trim Galore pipeline and Unicycler genome assembly. Whole-genome sequence reads and assemblies were submitted to NCBI under BioProject PRJNA1126273, containing BioSamples SAMN41936528 through SAMN41936546.

### 2.4. Bioinformatics

NCBI genome assemblies were downloaded 10 January 2024, using genome_updater v0.6.3 (https://github.com/pirovc/genome_updater, accessed on 10 January 2024). Phylogenomic trees of genome assemblies were produced based on kmer comparisons using PopPUNK v2.6.0 [41], based on core genome single nucleotide polymorphisms (SNPs) using ParSNP v1.7.4 [42], based on average nucleotide identity using pyANI v 0.2.12 [43], and based on percentage of conserved proteins using POCP-nf v 2.3.2 [44]. Newick trees were midpoint rooted using Archeopteryx v 0.9928 beta [45] and rendered in MicroReact [46]. Proksee was used for genome comparisons, FastANI analyses, and mobile element identification [47] based on the mobnileOG-db [48]. BLAST searches were employed at NCBI or a local installation of version 2.13.0 [49]. Assemblies were annotated using Prokka v1.14.5 [50]. Pan and core gene partitioning was performed using Roary v3.13.0 [51] and Pirate [52]. Scoary v1.6.16 [53] was used for pan-genome-wide association of output from Roary. Maast v1.0.8 [54] was used to generate a vcf containing genotype data for core genome SNPs for each genome, which were then processed further in Microsoft Office 365 Excel to calculate and plot SNP frequencies with respect to isolate traits. SeqKit [55] was used to convert fasta files to and from tab-delimited files, and to extract specific polypeptide sequences from the Prokka annotation files. Microsoft Excel was used to curate tab delimited sequences to filter for evident orthologs based on amino and terminal sequences, and for full-length orthologs. Excel was also used to analyze and summarize BLAST and SNP data. Variant identification in DNA or polypeptide sequences was based on alignment using Clustal Omega in MegAlignPro (Lasergene v17; DNAStar, Inc.; Madison, WI, USA) with variant tables exported to Excel for filtering, sorting, tabulation and graphing.

## 3. Results

### 3.1. Phylogenomic Analyses Identify Three Lineages

The 35 presumptive isolates of *S. cohnii* from Experiments 29 (Fall 2020) and 30 (Spring 2021) had been assigned to species based on PCR sequencing of the 16S V1-V5 region [19,36]. Species was verified by qPCR-HRM (high resolution melt) with primers specific for the *S. cohnii* pfbA gene [38], then each isolate was scored for color on CO and CS chromogenic media. Four different color types, designated A through D, were identified (Table 1). We submitted 18 genomic DNAs representing all four color types for Illumina sequencing. Unicycler assemblies of the trimmed sequence reads ranged from 20–40 contigs, with a consistent size of 2.58–2.70 Mbp (Appendix A), and GC percentage ranging from 32.31 to 32.53. These values agree with those for most *S. cohnii* assemblies in NCBI.

We downloaded all 96 assemblies from NCBI for *S. cohnii* and all 16 assemblies for *Staphylococcus urealyticus* (sometimes mis-labeled as *S. ureilyticus*). Lavecchi et al. [33] proposed that *S. cohnii* and *S. urealyticus* actually represent a single species complex. Our initial evaluation of the assemblies showed that 108 of the 112 assemblies from NCBI were of sufficient quality and sequence relatedness. Four assemblies were excluded because they were either misidentified for species, duplicate assemblies, or bad assemblies of mixed species (Appendix A). We used four different methods to examine the phylogenomic relationships of the 108 NCBI assemblies together with our 18 new assemblies. Figure 1 shows the PopPUNK tree based on core genome kmer comparisons. Appendix A is the ParSNP tree based on core genome SNPs. Appendix A is a neighbor joining tree of the pyANI pair-wise average nucleotide identity (ANI) distances. Appendix A is a neighbor-joining tree based on distances calculated from Percentage of Conserved Proteins (POCP). PopPUNK, ParSNP, and pyANI produce very similar trees showing three distinct lineages. These results are similar to the findings of Lavecchia et al. based on core genome multilocus alignments for 65 assemblies. Our Lineage 1, 2, and 3 correspond to their clades B, A2, and A1, respectively. The *S. urealyticus* assemblies (circles) are restricted to Lineage 2. BLASTn alignments of the 16S V1-V5 regions show that all 124 assemblies have the same 16S rDNA genes with a few exceptions for polymorphisms, likely attributable to assembly artifacts. Five isolates, DE0361, DE0360, SE4-2, SE4-1, and SE3-10, consistently appear intermediate between Lineage 3 and the other two lineages in all three trees. Isolate hu-01 appears intermediate only in the ParSNP and pyANI trees. Excluding the five intermediate isolates, the average ANI distance from Lineage 3 to either Lineage 1 or 2 is less than the distance from Lineage 1 to Lineage 2 (Table 2). The genetic distance between lineages is 8 to 10 times the genetic distance within each lineage. Thus, Lineages 1 and 2 are the most different with Lineage 3 in-between. The five intermediate isolates (m) are closer to Lineage 3 and slightly closer to Lineage 2 than Lineage 1.

POCP is a generally accepted method for genus and species delineation [56]. The POCP based tree in Appendix A does not show three lineages and the topology is quite distinct from the three other methods. Particularly striking is the placement on long branches of *S. urealyticus* 3636 and *S. cohnii* SNUC-4556. The assembly for isolate 3636 comprises 2.5 Mb in 1291 contigs with an N50 of only 3.6 kbp. Thus, the overall small contig size likely results in fewer and shorter predicted polypeptides, and thus a much lower POCP value. The assembly for isolate SNUC_4556 consists of 1461 contigs spanning 3.2 Mb with a N50 of 8.4 kb. The genome length suggests contamination and the large number of short contigs would also compromise the POCP value. The application of POCP for comparisons of a large number of draft assemblies appears to explain the radically different phylogeny in Appendix A and argues for only applying POCP to complete genome assemblies. Table 3 presents POCP and ANI values comparing the nine complete or chromosome-level assemblies (four each from lineages 1 and 3, and one from Lineage 2) along with four complete genomes for *Staphylococcus nepalensis* as the closest relative to *S. cohnii/S. urealyticus* [57]. POCP values within *S. nepalensis* or within lineages range from 90–95 and between lineages from 88–89, while each lineage is more distant at 84–85 compared to *S. nepalensis.* ANI values present the same picture of each lineage being approximately the same distance from *S. nepalensis* with less distance between lineages.

FastANI comparisons of the nine complete genomes along with those from *S. nepalensis* indicate that the genomes in all three lineages and both species are largely syntenic (Appendix A). However, the average genome size is larger for *S. nepalensis* at 2.9 Mbp versus *S. cohnii/S. urealyticus* at 2.5 Mbp. The analyses are complicated by not all assemblies starting at the same point or even in the same orientation. For example, the FDAARGOS_538 genome was best compared as a reverse complement, while the FDAARGOS_334 genome is circularly permuted relative to the representatives from lineages 2 and 3 and *S. nepalensis*.

### 3.2. Lineage Association with Host and Virulence

Examination of the hosts and disease status of the isolates in each lineage suggest that Lineage 3 may be largely non-pathogenic. The lineage contains a number of environmental (black) and plant (gray) isolates. Unfortunately, many of the isolates from mammals and humans have NCBI biosample data that are largely inconclusive for disease status. Human isolates (green) appear to be restricted to Lineage 1 and 3. Summaries of the biosample data for the human-associated isolates in Lineage 3 include:YNSA55: China, human, survey of S. aureus, no disease state recordedNCTC11041: Homo sapiens skin, no diseaseFDAARGOS-538: Homo sapiens, clinical isolateH62: air, environmentalAcroc: human skin, no disease state recordedRD01, RD04, RD09: Bacteria were isolated from multiple surfaces in washrooms on a university campus in Alberta, Canada.Hu-01: China Homo sapiens, skin swab, no diseaseG22B2: Homo sapiens, gall bladder

G22B2 and FDAARGOS-538 may be the only Lineage 3 isolates from diseased samples. The human-associated isolates in Lineage 1 include:57: Homo sapiens, blood; opportunistic pathogen isolated from human patient532: Homo sapiens, catheter; opportunistic pathogen isolated from human patientMUWRP0921: human UTIFDAARGOS_334: Homo sapiens, clinical isolateP3-SID1418: human skin, toe web spaceFDAARGOS_744: Homo sapiens, clinical isolateNBRC_109766: unknownRIT614: smartphoneNsq225: human, linezolid-resistant clinical isolateNsq226: human, linezolid-resistant clinical isolate3636: human oral infection

Therefore Lineage 1 appears to be more clearly associated with human pathogenesis. None of the 40 isolates in Lineage 2 are human isolates. The dairy isolates (red) are found only in Lineage 1 and 2, as are our new chicken isolates (light blue). The other mammalian isolates (light green; not human and not dairy) can be found in Lineages 2 and 3, with the single mammal isolate in Lineage 3 isolated from a dog ear canal, and therefore perhaps not a pathogen. This is all consistent with pathogenesis being prevalent in Lineage 1 and 2 and not in Lineage 3. The five or six intermediate isolates are also not associated with a disease state in the biosample entries.

### 3.3. Horizontal Transfer between Lineages

Overall frequencies of core genome SNPs for all genomes in the three lineages were determined using Maast with FDAARGOS_538, a complete assembly from Lineage 3, as the reference. SNPs were categorized as intergenic (*n* = 39,360), non-wobble (*n* = 174,466), or wobble (*n* = 23,029) based on the NCBI annotation for the FDAARGOS_538 genome. SNP frequencies for each of the three codon categories were plotted for all three lineages based on chromosomal position in FDAARGOS_538. Figure 2 shows the plots for SNP frequency differences for non-wobble SNPs for pair-wise comparisons of the three lineages (panel A) and for each of the lineages compared to the five intermediate isolates (designated m in Appendix A). Plots for the other two SNP categories show very similar patterns. SNP frequency differences were averaged over a 40 SNP sliding window to smooth the graphs. Gaps in the plots represent mobile elements (transposons, prophage, chromosomal islands) and tRNA/rDNA clusters. The frequency difference plots shows that Lineage 1 genomes have a high SNP frequency difference from both Lineage 2 and Lineage 3 throughout the chromosome except for a 186 kbp region from 1977 to 2162 (region 4 in panel B) spanning 184 protein encoding genes. For that region, Lineage 1 and Lineage 3 have similar SNP frequencies and a high SNP frequency difference from Lineage 2. Thus, Lineage 3 appears to have arisen from Lineage 2 by the replacement of this chromosomal region with the orthologous region from Lineage 1. KEGG analysis of EC numbers and inspection of the annotations did not identify a particular pathway or virulence function for these 184 genes. Figure 2 panel B shows the intermediate isolates also contain region 4 from Lineage 1 but also appear to have acquired region 2 (482 to 837 kbp) from Lineage 1. There are also region 1 (42-to 482 kbp) and region 3 (837 to 866 kbp) that appear to be moderately divergent from all of Lineage 1, 2, and 3. BLASTn queries of the NCBI microbial genomes database (i.e., 19 May 2024 query of draft and complete genomes in Staphylococcaceae excluding both *S. cohnii* and *S. urealyticus/ureilyticus*) with subsequences from regions 1 and 3 did not identify any potential source of this region in other Staphylococcus species.

### 3.4. Lineage Specific Polypeptides

Lavecchia et al. [33] previously identified 24 “clade-specific” genes based on 66 genomes of *S. cohnii* and *S. urealyticus.* With an expanded set of 126 genomes, we readdressed this question. The Prokka generated predicted polypeptides (.faa) files for all 126 genomes were catenated and used to make a blast protein database, which was analyzed by BLASTp queries with the .faa file for reference genome from each lineage: Lineages 1 FDAARGOS_334, Lineage 2 SDAQ-1, and Lineage 3 FDAARGOS_538. BLASTp results were filtered for pident*qcov ≥ 8100 (i.e., average of pident and qcov ≥ 90), then evaluated for hits for all of the isolates in each lineage. This identified 36 polypeptides present in all genomes from Lineage 1 and missing in one, or both, of the other lineages (Appendix A). Seven, possibly eight, of the polypeptides missing from Lineage 3 but present in all Lineage 2 genomes are clustered from PEG (protein encoded gene) 540 to 570, including genes for Nitrate metabolism, nitrogen/oxygen sensors, production of uroporphyrinogen, and a helix-turn-helix regulatory protein.

There are six polypeptides that are unique to a single lineage. In Lineage 1, there are three polypeptides not found in Lineage 2 or 3, a 321 residue RNA binding protein, and a 38 residue hypothetical protein, although BLASTp scoring for a 38 residue oligo peptide can be questionable. There is only one polypeptide uniquely present in Lineage 2; a 67 residue hypothetical protein. Lineage 3 has three unique polypeptides; a 67 residue DUF1146 domain containing protein, a 221 residue metallo-beta-lactamase domain-containing protein, and a 42 residue IS family transposase. Inspection of the annotations reveals no obvious virulence determinants associated with the radiation of Lineages 1 and 2 from Lineage 3. Where Lavecchia et al. identified 24 clade-specific polypeptides, we identified 36 lineage-specific genes. Reciprocal BLASTp analyses of their 24 vs. the 37 from this work identify only three possible overlaps. They identified a 50 residue hypothetical protein in clade A1 (equivalent to our Lineage 3) that corresponds with a 38 residue IS1182 transposase in our data. They identified a 126 residue LytTR family transcriptional regulator in clade A2 (our Lineage 2) that is related (pident = 32; qcov = 62) to a 152 residue LytTR family transcriptional regulator that we found present in all of Lineage 1, some of Lineage 2, and none of Lineage 3. They identified a 281 residue DUF4352 domain-containing protein in clade B that is highly related (pident = 99; qcov 100) to a 321 residue RNA binding protein uniquely present in Lineage 1 (equivalent to their clade B). Although both groups used BLASTp to identify clade/lineage-specific polypeptides, the results are not concordant and may relate to annotations based on incomplete assemblies and the number of assemblies utilized in each analysis.

The Prokka annotations were processed further through Roary to identify core and accessory genomes. The Roary output was then analyzed using Scoary to identify accessory genes positively or negatively associated with specific traits. Traits included host (i.e., human, mammal, animal, chicken) and lineage. However, there is always concern with assigning host from NCBI Biosample data. For example, isolates from washroom sinks, or from human skin swabs, could be from any source (i.e., plants, soil) or they could be valid human associated strains. The Roary-Scoary output identified no host-specific accessory genes (100% for both Specificity and Sensitivity). Roary identified 1017 core genes (99–100% of strains) with an additional 216 soft core genes (≥95%). Shell genes (≥15% and <95%) numbered 23,274, with 10,269 cloud genes (>0% and <15%). Conversely, Pirate uses a variable percentage identity threshold to partition clusters into pan and core genome with 2076 genes in ≥95% of isolates; 446 in ≥25% and <95%; with 4791 < 25%.

### 3.5. Lineage Diagnostic qPCR

The Roary-Scoary pipeline did identify lineage specific genes (Table 4). Lineage 1 appeared to have the highest number of lineage-specific genes with 84, even though there were more genomes (Table 2) included from that lineage. Further inspection of these lineage-specific genes reveals that most actually have orthologs in the other lineages where the orthologs differ by enough SNPs and indels that Roary identifies them as different genes when run with the default settings. Based on the Scoary output, we selected the Arylamine-N-acetyltransferase (nat) and GTPase (der) for design of lineage specific PCR primers. The primers targeted multiple SNP positions specific to the ortholog of a particular lineage. DerF1 + R1 was specific for Lineage 1, NatF1 + R1 specific for Lineage 3, and NatF1 + R1 specific for Lineage 2. These primers were employed in qPCR-HRM to diagnosis the remaining 19 *S. cohnii* isolates from Experiment 29 and 30 for which we did not have genome assemblies (Table 1). The qPCR-HRM results were clear for 18 of the 19 additional isolates but isolate 1645 was positive for DerF1 + R1 and NatF2 + R2, suggesting it was both Lineage 1 and Lineage 2. Therefore, we sequenced and assembled the genome for 1645 and based on PopPUNK analysis 1645 clusters with 1701 and 1649, which appear as an early branch of Lineage 1 (Figure 1). The Prokka annotation, as well as the Roary-Scoary analyses, identify 1649 and 1701 as containing both the Lineage 1 versions of der and nat. ANI computed distances for 1645 to 1701 is 0.9893; 1645 to 1649 is 0.9890; and 1649 to 1701 is 0.9990. The highest value for any of these three to the rest of Lineage 1 is less than 0.97. Prokka annotated contigs containing the Nat gene from 1645 (contig3 263 kbp), 1649 (contig2 427 kbp), and 1701 (contig3 340 kbp) were aligned in MegAlignPro along with a sequence variation plot (Appendix A). The contigs differ for annotation and sequence up to the RlmH gene near 23.9 kbp in contig3 from 1645. The three contigs then appear syntenic through the end of all three contigs (approximately 239 kbp); however, for the last approximately 50 kbp, the 1645 contig has an abundant number of SNPs and indels relative to the contigs from 1649 and 1701. The sequence variation begins two genes upstream of the nat gene. The alignments and sequence variation are most consistent with 1645 having acquired an approximately 50 kbp region, including the nat gene, from a Lineage 2 isolate. Whether this was by transformation, conjugation, or transduction is not known but is additional evidence of horizontal transfer between lineages. Further investigation would necessitate complete chromosome level assemblies of these three isolates for more extensive comparisons.

With the diagnosis of lineage for all 35 *S. cohnii* isolates, we determined that colony color on CO and CS media was not always a good indicator of lineage (Table 1). Although pink on CO and turquoise on CS, which was exclusively Lineage 1 (7 of 7), and white on CO and light green on CS was exclusively Lineage 2 (5 of 5). For pink on CO and light green on CS, two were Lineage 1 and three were Lineage 2. For turquoise on both CO and CS, sixteen isolates were Lineage 1 and two were Lineage 2.

### 3.6. S. cohnii Plasmids

There are sixteen plasmids annotated in the five complete assemblies (five in FDAARGOS_334, one in FDAARGOS_538, six in FDAARGOS_744, two in SDAQ-1, and two in SNUDS2). These were used as BLASTn queries to screen for highly related sequences in all the assemblies. The work was primarily focused on plasmid association with particular lineages or hosts (Appendix A). Our subjective measure of prevalence was 50% or greater for the maximum BLASTn bitscore for the plasmid with itself. The three largest (29–37 kbp) plasmids in FDAARGOS_334 are partially related to each other, as well as to the 34 kbp plasmid in FDAARGOS_538, the four largest (36–53 kbp) plasmids in FDAARGOS_744, plasmid 2 in SDAQ-1, and plasmid 1 in SNUDS-2. However, Clustal Omega alignments of these eight plasmids show that the relationships are piecemeal and not shared across all the plasmids (Appendix A). Related sequences based on BLASTn bitscores can be found in other assemblies but are limited to 19 or fewer genomes depending on the plasmid used in the query. There were no host-specific or lineage-specific-associations for any of the 16 plasmids.

### 3.7. Genetic Diversity in BCO Infections

Both Experiment 29 and Experiment 30 yielded a high degree of genetic diversity in *S. cohnii* genomes, as both lineages were isolated from each experiment (Figure 1; Experiment 29 gold star and Experiment 30 green plus). In Experiment 29, there were four closely related Lineage 2 isolates and six Lineage 1 isolates from four different radiations. For Experiment 30, there were isolates closely related to some of the diverse isolates in Experiment 29 (Figure 1, see 1711 and 1701) but wholly new genotypes were also present in Experiment 30 (e.g., 1697, 1714, and 1715). When we assigned lineage using qPCR-HRM for the rest of the isolates, we found Lineage 1 predominated in Experiment 29 (24 of 28 isolates) while Lineage 2 predominated in Experiment 30 (6 of 7 isolates). The genetic diversity observed within experiments also occurs within individual birds. Isolates 1659 and 1664 were from bird 3B in Experiment 29 and map to Lineage 1 and 2, respectively. Similarly, 1661 and 1665 are from bird 9A in Experiment 29 and map to Lineage 1 and 2, respectively. Further, Bird 10A in Experiment 29 yielded isolates 1637 and 1649, which are both Lineage 1 but in quite distinct radiations within that Lineage (Figure 1).

## 4. Discussion

Classic bacteriological methods (e.g., colony morphology, enzyme profiles, sugar fermentation, fatty-acid profiles, DNA-DNA hybridization) led others to propose that *S. cohnii* from humans and primates consisted of two distinguishable sub species, *S. cohnii* subsp. *cohnii* and *S. cohnii* subsp. *urealyticus* (see [32]). More recently, based on alignments of 1468 core genes from 66 genomes, others suggested that the *S. cohnii* complex (*S. cohnii*, *S. cohnii* subsp. *cohnii,* and *S. cohnii* subsp. *urealyticus*) comprises at least three different clades [33]. Their clades, A1, A2, and B, correspond to our lineages, 3, 2, and 1, respectively. The tree structure produced by Lavecchia et al. suggested a closer alliance between A1 and A2, which is in agreement with our ANI data showing Lineage 2 and Lineage 3 are closer to each other than either is to Lineage 1 (Table 2). Using three different methods for whole genome comparison, we derived the same basic topology of three lineages with a few intermediate isolates (Figure 1 and Appendix A). Typically, species delineation uses a threshold of genomic relatedness of 95–96% [43,58,59]. Table 2 shows average ANI values greater than 0.96 between Lineage 2 and 3, and just below 0.93 from both Lineage 2 and 3 to Lineage 1. Thus, it would appear that Lineage 1 might be considered as one species and Lineage 2 and 3 a different, single species. However, there are intermediate isolates we grouped into “m”, which have ANI values when compared to Lineages 1, 2, and 3 of 0.94, 0.95, and 0.98, respectively. Therefore, drawing strict species boundaries would be problematic. Further, as we found for isolate 1645, genomic exchange appears to be occurring between Lineage 1 and 2, and even Lineage 3 appears to have possibly arisen by transfer of a 186 kbp region from a Lineage 1 genome to replace the orthologous region in a Lineage 2 genome (Figure 2). Lineage m isolates also appear to be distinct from both Lineage 1, 2, and 3 for additional regions in the genome. Therefore, a single species complex may be more appropriate. With further investigation and production of additional complete genome assemblies, it may be possible to identify additional genomic transfers between the three lineages. The intermediate isolates we classified as “m” might be particularly appropriate genomes to examine. Transferred regions indicated from our data include: (a) at least 50 kbp acquired from Lineage 2 in isolate 1645, (b) 186 kbp acquired by a Lineage 2 genome from Lineage 1 to originate Lineage 3, (c) the 440 kbp region 1 and 29 kbp region 3 (Figure 2B) acquired by a Lineage 3 genome from an unknown source in the intermediate m genomes, and (d) 355 kbp region 2 (Figure 2B) identified in the intermediate m genomes acquired from a Lineage 1 genome. Given the large sizes, these regions are most likely acquired through conjugative transfer, rather than transduction or direct uptake of environmental DNA. Prophage mobilization of distant chromosomal elements has been documented in *Staphylococcus aureus* [60] but the transfers are limited to phage head capacity. Given that pathogenesis of staphylococci is often associated with biofilm formation we speculate that these horizontal transfers may be promulgated within biofilms.

Given the evidence of genetic exchanges between the lineages and the >92% relatedness measurements, our data is most supportive of designating all isolates of *S. urealyticus* and *S. cohnii* as members of a single species complex, similar to the complex comprising *Escherichia coli* and Shigella. Whole genome-based examination of *E. coli*/*Shigella* complex identifies fourteen phylogroups with four Shigella species clustered within *E. coli* [61,62]. Further pan-genome analyses of *E. coli* genomes suggested that major differences were associated with small mobile genome islands [63] or small sets of pan-genes [64]. There was no evidence for transfer of large chromosomal regions as we have detected in *S. cohnii* isolate 1645, Lineage 3, and the intermediate m genomes.

In our BCO induction experiments, source birds are raised on wire-flooring to stress the birds inducing lameness [7,8,28,31,36,65]. The infection spreads through the air to target birds raised on standard litter flooring, inducing BCO-lameness in 40–50% of the target birds. The necrotic lesions in the proximal femoral and tibial heads yield multiple different species even within a single bird. The original source of these bacteria could be the environment (the specific barn, farm personnel, insects, etc.). However, the more likely source is the inherent microbiome of the birds on wire-flooring. For both experiment 29 and 30, the same facility was used with no intervening flocks and the facility was cleaned out and thoroughly disinfected (water lines, floors and equipment) with dilute bleach between experiments. The source of the one-day-old chicks for both experiments was a commercial hatchery but the eggs were likely from different breeder flocks. The range of bacterial species identified from the BCO lesions differed between Experiment 29 and Experiment 30, but in both experiments the most numerous species was *S. cohnii*. Subsequent experiments performed in the same facility have detected far fewer isolates of *S. cohnii*. Since all the experiments were in the same research facility and sourced from the same hatchery, the breadth and genomic diversity of the bacteria present in the BCO lesions appears to vary with the batch of chicks and likely the particular breeder flock producing the eggs. Thus, our data from the genomic diversity of *S. cohnii* isolates suggests that even for 1-day old chicks they must have a rather complex and diverse microbiome.

We have developed inexpensive and rapid methods for environmental sampling for species identification and quantification [37]. It will be important to determine the levels of different Staphylococci in the air and feces during our BCO challenge experiments to understand the dynamics of the spread of pathogens (e.g., *S. aureus*, *S. cohnii*, *E. coli*, *Salmonella* sp.). Further examination must also be made of the microbiome of chicks at hatch to determine the contributions of the parent flock vs. the hatchery environment.

## Figures and Tables

**Figure 1 microorganisms-12-01549-f001:**
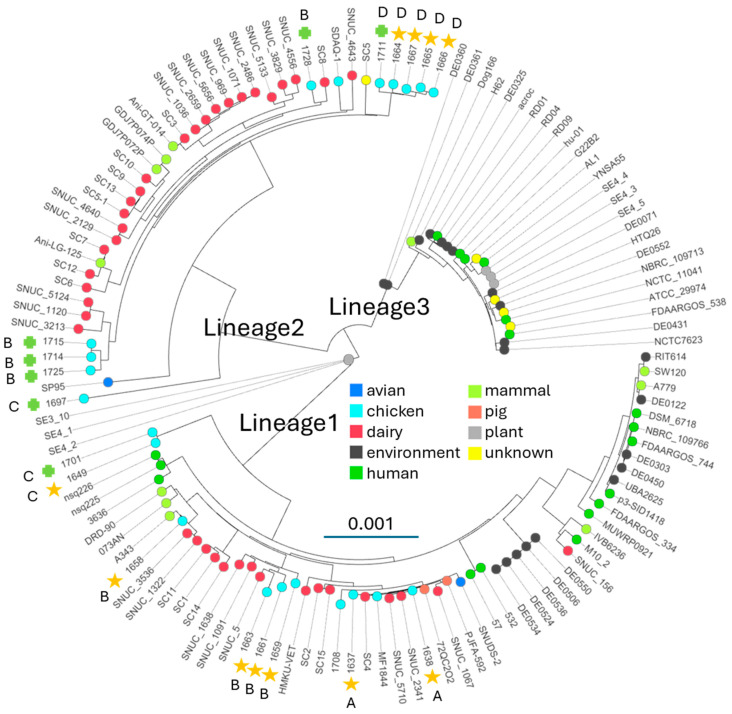
PopPUNK phylogenomic tree of genomes from 126 isolates of *Staphylococcus cohnii* (squares) and *Staphylococcus urealyticus* (*circles*). Strain names are the leaves, with nodes colored according to host (legend) as determined from NCBI Biosample entries. Chicken isolates from Experiment 29 (gold star) and Experiment 30 (green plus) are indicated along with color group (A, B, C, D) based on growth on CHROMagar media (Table 1). The branches for the three lineages described in the text are indicated.

**Figure 2 microorganisms-12-01549-f002:**
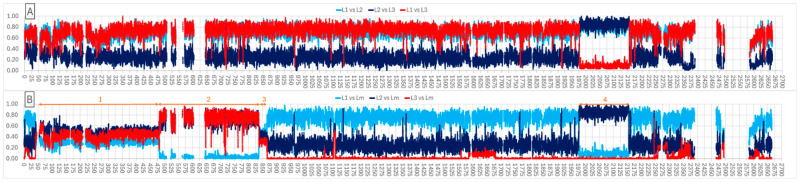
Core genome SNP frequency plots comparing frequencies of non-wobble SNPs in three lineages of *Staphylococcus cohnii* and *Staphylococcus urealyticus*. FDAARGOS_538 was the reference for SNP annotation and chromosomal position (X axis in kbp). Panel A shows the difference in SNP frequencies between Lineage 1 and Lineage 2 (L1 vs. L2; light blue), Lineage 2 and Lineage 3 (L2 vs. L3; dark blue), and Lineage 1 and Lineage 3 (L1 vs. L3; red). Panel B plots the non-wobble SNP frequency differences for the three Lineages relative to the five intermediate isolates (Lm) categorized as m in Appendix A. Lineage 1 and m (L1 vs. Lm; light blue), Lineage 2 and m (L2 vs. Lm; dark blue), and Lineage 3 and m (L3 vs. Lm; red). Regions 1 through 4 discussed in the text are delimited in Panel B.

**Table 1 microorganisms-12-01549-t001:** *Staphylococcus cohnii* isolates from Experiment (Exp) 29 and 30. For each Isolate Number is provided the source Bird identifier, whether whole genome sequence (WGS) was performed, the Lineage as determined from WGS or qPCR-HRM, and Colony Color on CHROMagar Orientaion (CO) or CHROMagar Staphylococcus (CS); l—lightgreen/lime, p—pink, t—turquoise, w—white.

Isolate					Colony Color	
Number	Exp	Bird	WGS	Lineage	CO	CS
1637	29	10A	yes	1	p	l
1649	29	10A	yes	1	t	t
1662	29	10B		1	p	l
1667	29	10B	yes	2	w	l
1670	29	10B		1	t	t
1638	29	11A	yes	1	p	l
1650	29	11A		1	t	t
1651	29	11B		1	t	t
1652	29	12B		1	t	t
1653	29	13A		1	t	t
1654	29	13B		1	t	t
1663	29	13B	yes	1	p	l
1655	29	14A		1	t	t
1656	29	14B		1	t	t
1657	29	2A		1	t	t
1658	29	2A	yes	1	p	l
1659	29	3B	yes	1	p	l
1664	29	3B	yes	2	w	l
1644	29	5B		1	t	t
1645	29	6A	yes	1	t	t
1636	29	6B		1	p	l
1646	29	6B		1	t	t
1661	29	9A	yes	1	p	l
1665	29	9A	yes	2	w	l
1669	29	9A		1	t	t
1639	29	9B		1	p	l
1648	29	9B		1	t	t
1666	29	9B	yes	2	w	l
1697	30	2	yes	2	t	t
1725	30	3	yes	2	t	t
1711	30	4	yes	2	w	l
1701	30	6	yes	1	t	t
1728	30	8	yes	2	p	l
1714	30	10	yes	2	p	l
1715	30	10	yes	2	p	l

**Table 2 microorganisms-12-01549-t002:** Average Nucleotide Identity (±SD) between the isolates of the three lineages plus the intermediate (m) isolates.

Lineage	n	1	2	3	m
1	56	0.99012 ± 0.00771	0.92023 ± 0.00566	0.92526 ± 0.00333	0.94591 ± 0.00196
2	40		0.99290 ± 0.00591	0.96082 ± 0.00171	0.94935 ± 0.00025
3	23			0.99558 ± 0.00368	0.97667 ± 0.00216
m	5				0.99996 ± 0.00004

**Table 3 microorganisms-12-01549-t003:** Percentage of Conserved Proteins (POCP) and Average Nucleotide Identity (ANI) values (±SD) for complete genomes for *Staphylococcus nepalensis* (Sn) and the three lineages of *Staphylococcus cohnii* (1, 2, 3) discussed in the text. Values for Lineage 2 compared to Lineage 2 are not applicable (NA) as there is only one complete genome for that lineage.

		Sn	1	2	3
	n	4	4	1	4
POCP	Sn	93.9 ± 1.8	84.2 ± 1.4	85.4 ± 1.6	84.0 ± 0.7
	1		90.4 ± 2.3	88.5 ± 1.5	89.2 ± 1.6
	2			NA	88.2 ± 0.6
	3				95.3 ± 2.0
ANI	Sn	0.997 ± 0.0015	0.857 ± 0.0007	0.879 ± 0.0459	0.859 ± 0.0012
	1		0.990 ± 0.0036	0.919 ± 0.0003	0.925 ± 0.0015
	2			NA	0.961 ± 0.0004
	3				0.997 ± 0.0017

**Table 4 microorganisms-12-01549-t004:** Scoary output for genes differentially represented in *Staphylococcus cohnii* genomes corresponding to particular traits. Gene counts are for those showing statistical differences for the phenotype (Bonferroni corrected *p* < 0.05), for those with positive association (Odds Ratio > 4.4), negative association (Odds Ratio < 0.24), and those with 100% Specificity (presence in all isolates with the trait), and Sensitivity (present only in isolates with the trait).

Trait	Gene Count			
	Bonferroni *p* < 0.05	Odds Ratio		100% Specificity and Sensitivity
		>4.4	<0.24	
Lineage 1	1756	857	899	84
Lineage 2	1716	867	849	14
Lineage 3	1311	707	604	13
Human	333	195	138	0
Mammal	7	0	7	0
Avian	7	7	0	0
Chicken	17	17	0	0
Animal	166	37	129	0

## Data Availability

All genomes analyzed in this research are available from the National Center for Biotechnology Information, National Library of Medicine (https://www.ncbi.nlm.nih.gov).

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
