# Peer review of "Phylogenomic Analyses of Three Distinct Lineages Uniting Staphylococcus cohnii and Staphylococcus urealyticus from Diverse Hosts"

_microorganisms, 2024, doi:10.3390/microorganisms12081549_

Round 1

Reviewer 1 Report

Comments and Suggestions for Authors

This study analyzed three different lineages of Staphylococcus cohnii and Staphylococcus urealyticum from different hosts. It expands the diversity of Staphylococci associated with osteomyelitis in poultry, showing that the microbiomes of Staphylococcus cohnii and Staphylococcus urealyticum are highly diverse. Determining the levels of different Staphylococci in the air and feces in BCO challenge experiments is important to understand the dynamics of pathogen transmission. The conclusion in this manuscript is reasonable.

I don’t know why the supplemental data in WORD file is hard for me to read. It is too dark.

 The background of S. cohnii and Staphylococcus urealyticus in Introduction seems simple and not enough. Can you please add more in this section?

In method section: Lack the experiment for inducing lameness.

There are revisions may need:

1.    Line 99: What is the anneal temperature?

2.    Line 108: Is the anneal temperature only 10 oC.?

3.    Line 119: RNAse usually presented as RNase.

4.    Line 145: “curate .tab” is there any typo error for the dot? Or extra space?

Author Response

I don’t know why the supplemental data in WORD file is hard for me to read. It is too dark. Response: We checked the Supplemental Figures.docx in the manuscript-supplemntary.zip file from the publisher and there was no problem viewing the document in Word.

 The background of S. cohnii and Staphylococcus urealyticus in Introduction seems simple and not enough. Can you please add more in this section? Response: As we state in the introduction there is no literature on S. cohnii in chickens, and there are only a few references about the genomic diversity in human isolates. We do cite the most relevant articles (see lines 61-66).

In method section: Lack the experiment for inducing lameness. Response: We provided the citation to the experiments for inducing lameness as they involved additional testing and constitute a different set of analyses and results. Line 82 states: “Details of the animal work have been published elsewhere [31].”

There are revisions may need:

  1. Line 99: What is the anneal temperature? Response: the anneal temperatures are given below where the primer sequences are provided. However, we added “{see below)” for clarity.
  2. Line 108: Is the anneal temperature only 10 oC.? Response:This has been corrected
  3. Line 119: RNAse usually presented as RNase. Response: corrected
  4. Line 145: “curate .tab” is there any typo error for the dot? Or extra space? Response: this is not a typo. .tab-formatted sequences is when fasta files are converted to a tab-delimited table. So we have made this less “jargon” based.

Reviewer 2 Report

Comments and Suggestions for Authors

This paper examines the relatedness of a set of about a hundred Staphylococcus cohnii and S. ureilyticus genomes.  The methods are well described, and results are clear - basically three subspecies, in agreement with other authors. More extensive comparison between the three lineages revealed specific genes for each lineage, and it appears that perhaps lineage 3 might have come about as the result of a transfer of a 186,000 bp region from lineage 1 to lineage 2, creating lineage 3.

This is interesting, but lineage 3 is considered more environmental, mainly found in enviornmental isolates, plants, and unknown origins. I'm confused as to how this relates to the initial purpose of the experiments, which was to look for potential causes of Bacterial Chondronecrosis with Osteomyelitis.  Perhaps this could be fleshed out a bit more in the discussion?

on page 7, line 230, the authors state: 

"The [FastANI] analyses are complicated by not all assemblies starting at the same point or even in the same orientation."

--> It is my understanding that FastANI works by taking both genomes and dividing them into small chunks of about 10,000 bp, and looking at the % identity (ANI) - but the program takes into account reverse compliments, so in principle it should not matter where the contigs start, nor their orientation.

Author Response

This paper examines the relatedness of a set of about a hundred Staphylococcus cohnii and S. ureilyticus genomes.  The methods are well described, and results are clear - basically three subspecies, in agreement with other authors. More extensive comparison between the three lineages revealed specific genes for each lineage, and it appears that perhaps lineage 3 might have come about as the result of a transfer of a 186,000 bp region from lineage 1 to lineage 2, creating lineage 3.

This is interesting, but lineage 3 is considered more environmental, mainly found in enviornmental isolates, plants, and unknown origins. I'm confused as to how this relates to the initial purpose of the experiments, which was to look for potential causes of Bacterial Chondronecrosis with Osteomyelitis.  Perhaps this could be fleshed out a bit more in the discussion?

Response: The purpose was not to look for potential causes of BCO. As we stated in the Introduction (Lines 66-73): “Given the paucity of publications, or genome assemblies, for S. cohnii infecting chickens, and the high-frequency of isolation in two successive BCO challenge experiments, we chose to examine the genomic diversity of our BCO isolates of S. cohnii. Standard microbiological techniques suggested a diverse population, and whole genome comparisons expand the available genomes for elucidating the evolution of this taxonomic group, and our understanding of the phylogenetic evolution of pathogenesis and host tropism. Our results also contribute to a better understanding of the etiology of BCO outbreaks in the broiler industry.” Therefore this paper is about the additional genomes, the host diversity, and the interchange of genomic segments in a species complex.  Given that others have tried to call Lineage 1 S. cohnii and Lineage 2  S. urealyticus our paper is an attempt to say there is much more to the story.

on page 7, line 230, the authors state: 

"The [FastANI] analyses are complicated by not all assemblies starting at the same point or even in the same orientation."

--> It is my understanding that FastANI works by taking both genomes and dividing them into small chunks of about 10,000 bp, and looking at the % identity (ANI) - but the program takes into account reverse compliments, so in principle it should not matter where the contigs start, nor their orientation.

Response: pyANI does a kmer comparison to determine an overall Average Nucleotide Identity score which would not be complicated by genomes starting at different places or being inverted. However FastANI analysis is a kmer based alignment as shown in Figure S4.  Since this produces a graph of the whole genome alignments there can be confusion if the genome sequences begin at different locations or are inverted.

Reviewer 3 Report

Comments and Suggestions for Authors

Phylogenomic analyses of three distinct lineages uniting Staphylococcus cohnii and Staphylococcus urealyticus from diverse hosts

 Thanks for work.

  1. The aim of the study should be specified at the end of the introduction section.
  2. Photos of joint lesions ‘’femoral head necrosis’’ could be added.
  3. A reference for the colour classification of the isolates (A to F) after culturing on CO and CS chromogenic media should be added.
  4. No references number (lines 155, 156, 217) or references results should be present in the results section (line 196). They should be transferred to the discussion section.
  5. In the discussion, no table’s (line 442) or figures (lines 444, 463) numbers should be mentioned.
  6. The ‘’detailed’’ results should not be repeated in the discussion.
  7. There is a problem in the abbreviation and the method of writing bacteria all over the manuscript. The name should be written in full and followed by abbreviation at the first mention, then written as abbreviation all over the manuscript. For example, line 41, Staphylococcus aureus should be followed by the abbreviation ‘’S. aureus’’. The same, in line 42, Staphylococcus agnetis should be followed by the abbreviation ‘’S. agnetis’’ and line 473, Escherichia coli. However, in line 61, Staphylococcus cohnii should be written as ‘’S. cohni’’.
  8. The repeated terms ‘’we, ours’’ could be replaced by ‘’the authors, this research, etc.’’.
  9. The pfbA and other genes should be written in a correct way.

Best wishes

Author Response

  1. The aim of the study should be specified at the end of the introduction section. Response: the aim of the study is presented in Lines 66-69, followed by significance of the work lines 69-73
  2. Photos of joint lesions ‘’femoral head necrosis’’ could be added. Response: Dr. Wideman has already published exquisite photographs of BCO lesions, as well as our co-author Dr. Alrubaye. The work in this manuscript is about the genomic diversity in cohnii and less about lameness.  This genomics work will be followed by manuscripts on the airborne infection and chick microbiome.
  3. A reference for the colour classification of the isolates (A to F) after culturing on CO and CS chromogenic media should be added. Response: we are not certain what sort of reference the reviewer wants. The CO an CS media are listed in the methods with the supplier.  The suppliers website has examples of colors.  We are not aware of any publication where cohnii was tested on CO or CS plates.
  4. No references number (lines 155, 156, 217) or references results should be present in the results section (line 196). They should be transferred to the discussion section. Response: We are not aware of any journal restriction on having citations in the results section.
  5. In the discussion, no table’s (line 442) or figures (lines 444, 463) numbers should be mentioned. Response: We are not aware of any journal restriction or style manual restricting references to tables and figures appearing in the discussion section.
  6. The ‘’detailed’’ results should not be repeated in the discussion. Response: with all due respect to the reviewer, many readers prefer an overview of the results emphasizing the highlights
  7. There is a problem in the abbreviation and the method of writing bacteria all over the manuscript. The name should be written in full and followed by abbreviation at the first mention, then written as abbreviation all over the manuscript. For example, line 41, Staphylococcus aureusshould be followed by the abbreviation ‘’S. aureus’’. The same, in line 42, Staphylococcus agnetis should be followed by the abbreviation ‘’ agnetis’’ and line 473, Escherichia coli. However, in line 61, Staphylococcus cohnii should be written as ‘’S. cohni’’. Response: we have followed the accepted practice of spelling out the genus in the first instance and then abbreviating the genus for any subsequent listings.  However, this process is separately followed for the article title, the abstract, and the article.  In addition figures and tables should be self-standing so the genus is presented in full.  We will work with the journal editor to follow the journals policy.
  8. The repeated terms ‘’we, ours’’ could be replaced by ‘’the authors, this research, etc.’’. Response: We agree that they could be replaced
  9. The pfbA and other genesshould be written in a correct way. Response: pfbA appears twice in the manuscript and is presented as pfbA in both instances. We have changed Der to der throughout with the exception of the primers DerF1+R1. We have done the same for Nat/nat